# Determinants of Complementary Feeding Indicators: A Secondary Analysis of Thailand Multiple Indicators Cluster Survey 2019

**DOI:** 10.3390/nu14204370

**Published:** 2022-10-18

**Authors:** Abhirat Supthanasup, Nisachol Cetthakrikul, Matthew Kelly, Haribondhu Sarma, Cathy Banwell

**Affiliations:** 1School of Human Ecology, Sukhothai Thammathirat Open University, Nonthaburi 11120, Thailand; 2International Health Policy Program, Ministry of Public Health, Nonthaburi 11000, Thailand; 3National Centre for Epidemiology and Population Health, Australian National University, Canberra, ACT 2601, Australia

**Keywords:** complementary feeding, determinants, infant feeding, Thailand, young child

## Abstract

Child complementary feeding (CF) practices meet dietary recommendations more often among educated, high-income groups. Much of the evidence for this association addresses inadequate CF for addressing child undernutrition. However, in many countries, including Thailand, child malnutrition assessments must now address under- and over-nutrition. More comprehensive data is needed to understand this complex situation. This study uses data from the Thailand Multiple Indicators Survey 2019, to identify the determinants of CF practices among 6–23-month children (*n* = 4125) using the newly developed WHO indicators. Logistic regression analysis was used to measure associations between sociodemographic factors and CF practices. In a fully adjusted model, child age, primary caregivers’ education, and household incomes were statistically associated with (in)appropriate CF practices. Older children aged 9–23 months, not only have better minimum dietary diversity (MDD), minimum acceptable diet (MAD), and egg and/or flesh food consumption (EFF), but also tend to consume more unhealthy foods. The proportion of inappropriate CF practices was higher among children living with caregivers other than their mothers. While maternal education and household income were positively associated with MDD and MAD, children of mothers from middle-class households consumed more sweetened beverages. Therefore, nutrition programs addressing different feeding problems should be developed specifically for different primary caregiver and demographic groups.

## 1. Introduction

Child malnutrition remains an alarming global issue, despite the remarkable progress in improving social and economic development [1]. Malnutrition, which is defined as “deficiencies, excesses, or imbalances in a person’s intake of energy and/or nutrients”, addresses two broad groups of conditions- undernutrition and overnutrition [2]. International organisations estimate that global stunting and wasting prevalence among children under five declined from 33.1% and 7.5% in 2000 to 22.0% and 6.7% in 2020, respectively [3]. However, the prevalence of overweight among children under five increased from 5.4% in 2000 to 5.7% in 2020 [3]. The prevalence of the two malnourished states displays significant variation across contexts and settings.

Child undernutrition has the greatest impact in low-and middle-income countries (LMICs). An analysis using national survey data from 50 LMICs showed a moderate decline in child stunting prevalence from 40.1% and 23.3% in 2000 to 31.9% and 17.5% in 2015, respectively [4]. Moreover, undernutrition has the greatest impact on Asian and African children. Approximately, 79 million Asian children younger than five years are affected by stunting and 31.9 million by wasting [3]. At the same time, the levels and trends of child overnutrition have been growing in many regions, especially Southeast Asia. In this region, the prevalence increased from only 3.7% in 2000 to 7.5% in 2020 [3]. Moreover, the impact of child overnutrition is greatest in upper-middle-income (8.8%) and high-income (7.8%) countries when compared with low-income (3.7%) and lower-middle-income (4.0%) countries [3].

In Thailand, a Multiple Indicator Cluster Survey (MICS) revealed that stunted, underweight, and wasted prevalence among young children had slightly improved between 2012 and 2019. The stunting rate decreased from 16.3% in 2012 to 10.5% in 2016 before slightly increasing to 13% in 2019 [5,6,7]. The trend in child underweight is similar to those in stunting, where the rate decreased from 9.2% in 2012 to 6.7% in 2016 and then rose to 8% in 2019. However, MICS shows a slight drop in levels of child wasting from 6.7% in 2012 to 5.4% in 2016, and then an increase to 8% in 2019. As well, like other upper-middle-income countries, Thailand is facing increasing levels of child overnutrition. Overweight prevalence among Thai children under five was under 2% in 1986 [8], it then steadily increased to 9% in the next 33 years [7]. Therefore, Thailand is now encountering a double burden of child malnutrition.

Inappropriate complementary feeding is a cause of malnutrition, both under and over nutrition, in children. Children being fed poor quality semi-solid food and/or with low meal frequency were associated with undernutrition, which includes three groups of conditions: stunting, wasting, and underweight [9,10]. Meanwhile, the early introduction of complementary feeding has been associated with the risk of early cessation of breastfeeding and the risk of overweight and obesity [11,12,13]. Caregivers, most often mothers, take responsibility for young child feeding, with their practices influencing child nutritional status. Child feeding practices are also influenced by multidimensional determinants, including personal, household and community factors. Previous studies tend to focus on determining the socioeconomic determinants of child feeding practices, especially in LMICs. Children that are older, and from better maternal education backgrounds, higher household wealth status and higher access to antenatal care are associated with better complementary feeding practices [14,15,16,17]. However, the determinants of child feeding practices vary in different contexts and settings.

In 2008, the WHO introduced indicators for assessing infant and young child feeding practices. These indicators were designed to assess trends and diversity in young child diets, identify children at risk of malnutrition and monitor progress in risk reduction, and evaluate interventions [18]. As the food landscape has changed, the WHO proposed a new set of indicators in 2021, covering dietary diversity, food groups, and unhealthy food and beverage consumption [19]. The summary of changes between the 2008 and 2021 set of indicators is described in Appendix A.

In Thailand, the determinants of child health and nutrition have undergone fundamental changes as the country has experienced rapid socio-demographic and economic change over the past decades. Thailand became an upper-middle-income country in 2011, and is experiencing the dual burdens of over and under nutrition. Consequently, gaining a better understanding of the current relationship between socioeconomic and demographic status and child feeding practices is needed. Therefore, the aim of this study is to use the newly developed WHO infant feeding indicators to assess patterns of complementary feeding among Thai children 6–23 months of age, the distribution of these patterns, and to identify the potential risk factors associated with inappropriate complementary feeding practices in Thailand.

## 2. Materials and Methods

This study is a cross-sectional quantitative analysis of secondary data. We analysed data for 4125 children aged 6–23 months obtained from the nationally representative Multiple Indicators Cluster Survey (MICS) 2019 to assess the association between complementary feeding patterns and child, caregivers, and other characteristics.

### 2.1. Data Sources and Variables

Multiple Indicators Cluster Survey (MICS) is an international cross-sectional household survey program developed and introduced by the United Nations Children’s Fund (UNICEF) to support countries to collect useful data about women and children. Countries can employ MICS to measure indicators or monitor the progress of Sustainable Development Goals. Furthermore, findings from MICS are able to be empirical evidence for developing policies to improve children’s diets [7].

This study analysed the Multiple Indicators Cluster Survey 2019 (MICS6), conducted by the National Statistical Office (NSO) in Thailand, between May and November 2019. MICS6 used face-to-face interviews by the trained field staff of NSO. NSO collected data from 17 of Thailand’s 77 provinces in five regions (Bangkok, Central, North, South, and Northeast). Field workers entered all data into tablets directly during the interviews. Each interview took around one hour. If the sampled respondent was not home or physically present during the first visit, NSO staff revisited at least three times [7].

Our study only used data from interviews with mothers or caregivers of children aged 6–23 months old. The main independent variables were child’s factors (gender and age), maternal/ caregivers’ factors (age, education, and nationality), and household and community level factors (household wealth index, residential areas, and geographical regions). The WHO complementary feeding indicators were dependent variables.

### 2.2. Complementary Feeding Indicators of WHO

We applied the new and updated infant and young child feeding indicators of the WHO [19]. Table 1 summarised the new WHO complementary feeding indicators.

### 2.3. Data Analysis

STATA software version 17 was used for all calculations (serial license number: 401709350741). There were three steps to the analysis. First, was a descriptive analysis to explain the characteristics of the sample and frequency of the complementary feeding indicators. Second, univariate logistic regression analysis was used to calculate the odds ratio (OR) to explore the association between complementary feeding indicators and each independent variable. Third, multivariate logistic regression was applied to find an adjusted odds ratio (AOR) to examine the association between the dependent variable and adjusted each independent variables, mutually adjusted for all other independent variables. We included all independent variables in the multivariate logistic regression analysis, even though they did not show statistically significant association in our analysis, because previous papers had suggested there may be an association. Statistical significance was measured at the 95% confidence level (*p*-value < 0.05).

## 3. Results

### 3.1. Characteristics of the Sample

The number of sampled children aged between 6–23 months was 4125. Male and female children were nearly equally represented in the sample, at 51.49% and 48.51% respectively. Most children were in the age range of 12–23 months (69.8%). The proportion of children cared for by people other than their mothers increased by the child’s age. Overall, around 30% of children were currently breastfed (reported receiving any breastmilk in the past 24 h). Moreover, around 38% of children with mothers as primary caregivers were currently being breastfed, while only 3% of those being cared for by others reported receiving breastmilk in the past 24 h. In terms of characteristics of mothers and other caregivers, about three-fourths of mothers were 20–35 years old. Approximately 96% of other caregivers were aged above 35. Mainly, mothers of these children hold a high school degree and above (86.57%), while the majority of other caregivers had only attained a primary level of education (75.40%). The majority of children, especially those being cared for by others, were in rural areas in the northeast of Thailand (Table 2).

### 3.2. Complementary Feeding Indicators

Table 3 presents the complementary feeding indicators of children being cared for by mothers and others across child age groups. Most children in the 6–8 months group had received solid, semi-solid, or soft foods (91.56% and 93.41% in groups being cared for by mothers and others, respectively). When comparing the proportions of other indicators between primary caregivers’ groups, there were significant differences in all indicators. Overall, less than half of 6–8 month old children meet minimum dietary diversity standards (MDD). There were higher proportions of children being cared for by mothers who achieved MDD than those with others (45.35% vs. 32.97%, *p* < 0.001 in the 6–8 months group; 65.95% vs. 61.38%, *p* = 0.009; 75.96% vs. 67.62%, *p* < 0.001 in the 12–17 months group; 74.09% vs. 67.54%, *p* < 0.001 in 18–23 months group). There were higher proportions of children being cared for by mothers achieving minimum acceptable diet standards (MAD) than those with others (42.19% vs. 27.27%, *p* < 0.001 in the 6–8 months group; 64.31% vs. 58.45%, *p* = 0.001 in the 9–11 months group; 70.39% vs.65.10%, *p* = 0.002 in the 12–17 months group; 70.37% vs. 63.68%, *p* < 0.001 in the 18–23 months group). Overall, most children aged 6–23 months had eggs or flesh meats and vegetables or fruit in the past 24 h, but the rates were lower for infants aged 6–8 months. The consumption rate of eggs or flesh meats and vegetables or fruit were lower in those children being cared for by others when compared to mothers. Furthermore, the consumption rate of sweet beverages and unhealthy foods increased as children aged. These consumption rates were higher in children being cared for by others when compared to mothers.

### 3.3. Determinants of Complementary Feeding Indicators

Findings of univariate logistic regression revealed that either living with mothers or not, the age of children had an association with almost all complementary feeding indicators, except, MMFF. For children who lived with mothers, sociodemographic factors (region or wealth) related to MDD, MMFF, MAD, EFF, SWB, UFC, and ZVF, while sociodemographic factors of children who lived with others associated with MDD, MAD, EFF, SWB, and ZVF. Education levels of primary caregivers had an association with some indicators in the ‘living with mothers’ group namely MDD, MMF, MAD, EFF, SWB, and ZVF, but the variable did not associate with the ‘living with others’ group. (Appendix A).

Table 4 presents the results of the multivariate logistic regression analysis in which all the main variables were included. The child’s age had strong associations with complementary feeding indicators. Older children tend to have better MDD, MMF, MAD, and EFF. After adjusting for covariates, children aged 12–17 months had strong associations with MDD in the group being cared for by mothers (adjusted odd ratios (AOR) = 4.35 [95% CI: 3.38–5.60]), and the group being cared for by others (4.75 [1.66–13.62]). There were associations between child age and EFF in children being cared for by both mothers and others; compared with those in youngest age group, those in aged 9–11 months (3.35 [2.32–4.85]) and (6.90 [1.62–29.31), 12–17 months (7.18 [4.89–10.54) and (18.95 [4.28–83.95]), and 18–23 months (9.54 [6.22–14.61]) and (84.02 [8.64–817.36]). Children of mothers with higher education and from wealthier households had better appropriate complementary feeding (MDD and MAD). Children of mothers with higher education backgrounds were more likely to achieve MDD (1.41 [1.10–1.81]) and MAD (1.41 [1.10–1.81]). Moreover, higher maternal education increased chances of achieving EFF. There was no association between other caregivers’ education and household wealth index, and EFF. However, better household incomes were associated with around three-fold increased change of achieving MAD in groups of children being cared for by others.

Conversely, older children in groups being cared for by mothers had strong associations with consuming unhealthy foods. Compared with children aged 6–8 months, those in the aged 9–11 months (1.87 [1.15–3.04]), 12–17 months (2.72 [1.75–4.23]), and 18–23 months (3.44 [2.20–5.37) had increased chance of unhealthy food consumption. Higher household income was a protective factor against feeding children unhealthy foods among those lived with mothers. Moreover, children of mothers from middle-class households and living in Bangkok and Central Thailand tend to consume sweet beverages. However, current breastfeeding was protective against sweet beverage consumption. Older children of mothers with better household income tend to consume more fruits and vegetables when compared with the youngest and lowest income groups. Further details are given in Table 5.

## 4. Discussion

This analysis of a nationally representative survey of Thai children, mothers and other caregivers reveals mixed patterns of both appropriate and inappropriate child feeding practices with some indicators at concerning levels. Overall, the complementary feeding practices of other caregivers were less favourable than those of mothers. Although we cannot identify who the non-mother caregivers are due to the limitation of MICS6 data their age range points to older generations. A study in the UK compared dietary provision between parents and grandparents found that parents scored higher for promoting balance and variety [20]. The results of this study also highlight the need of expanding interventions promoting healthier complementary feeding practices that target non-mother caregivers, including grandparents.

In this study, there were several factors that were associated with WHO complementary feeding indicators. Child age was regularly associated with inappropriate child feeding practices. Fewer children in the 6–8 months range, especially those being cared for by others (non-mothers), met the MDD compared with older age groups. Also, this group of children had a significantly lower MAD and lower EFF, and higher ZVF. Importantly, achieving MDD was also associated with consuming at least one egg or flesh food and at least one fruit or vegetable. It could point to an inappropriate diet composition for younger children, which predominantly consists of starchy food with little or no eggs, flesh foods, fruits and vegetables. This type of diet is considered low in nutrient density and with poor mineral availability [21], increasing the risk of malnutrition. In Thailand, traditional weaning foods are rice and banana [22]. A study of modern feeding practices that analysed recipes shared on Thai online peer support groups revealed that animal-source foods were not commonly fed until children were aged eight months and older due to allergy concerns [23]. However, a recent literature review points to the role of delayed introduction of allergenic food in increasing the risk for allergy development [24]. Moreover, a recent systematic review revealed that introducing a variety of vegetables at the early stage of weaning promotes vegetable acceptance [25]. Therefore, programs to improve complementary feeding practices need to focus on encouraging parents to feed their children eggs, flesh foods, and vegetables from the beginning of weaning process. 

While young Thai children still have challenges meeting dietary diversity needs, older children tend to consume more unhealthy foods. A systematic review of energy-dense, nutrient-poor foods highlighted the contribution that consumption of these types of food make to a substantial proportion of the energy intake among children younger than 23 months old in low- and middle-income countries [26]. While low dietary diversity is associated with child stunting in many settings [27,28], feeding young children with energy-dense foods is likely to increase the risk of overweight. Interestingly, early undernutrition followed by later overweight has promoted central adiposity and insulin resistance [29] which increase the risk of later developing non-communicable diseases [30].

It is often found that higher maternal education is associated with appropriate child feeding practices regarding to minimum dietary diversity, minimum meal frequencies, and minimum acceptable diet [15,16]. Our findings are in line with these previous studies in that children of mothers with higher educational level displayed better complementary feeding indicators (MDD and MAD). Also, a positive association was found between maternal education level and new feeding indicators, egg and/or flesh food consumption and vegetable and fruit consumption.

Findings from this study provide further support for associations between household income level and complementary feeding indicators on MDD and MAD. These results support the potential role of household incomes in providing children a wide variety of foods. For example, household income was associated with feeding diverse complementary foods in Nepal [31]. However, interestingly, feeding a variety of food to children might not be limited to healthy foods. In this study, children from middle-class families had higher sweet beverage consumption compared with the poorest and richest households. Geographical variation also plays a role in sweet beverage consumption.

Parents living in Bangkok and the central Thai region were more likely to provide their children with sugary beverages. There is limited evidence on associations between children’s sweet beverages consumption and household income and geography. However, urbanisation and rising household incomes provide middle-class families easy access to a variety of foods, including sweet beverages in accessible settings. These findings show that parental feeding practices often do not comply with the current guidelines. Recently, organizations including UNICEF and the European Society for Paediatric Gastroenterology, Hepatology, and Nutrition (ESPGHAN) published complementary feeding guidelines which recommended no provision of sugar-sweetened beverages [32,33]. Early life exposure to sweet beverages increases risks of obesity, dental caries, liver fat and non-alcoholic fatty liver disease (NAFLD) in later life [34,35,36]. Thus, information or other interventions are needed to encourage middle-class parents to avoid giving sweet beverages to their children. Interestingly, our findings pointed out that children who received breastmilk had a lower sweet beverage consumption. These findings align with a previous Brazilian study showing that breastfed children were less likely to consume sugary foods later in life [37]. 

### Strengths and Limitations

The strengths of this study include that it used generalizable national survey data. Therefore, sample groups represented the population in Thailand. Second, this study presents the association between determinants (individual and community), and complementary feeding indicators, of problems in child feeding in Thailand. Policymakers and relevant stakeholders can apply our findings to develop better complementary feeding policies and measures.

The limitations of this study are as follows; first, some populations who do not have a registered household number such as homeless people, illegal migrants, and people living in slum areas, were excluded from the survey because MICS6 selected households from the household registry from the Department of Provincial Administration. Second, MICS used a quantitative approach to collect data. Consequently, the perspectives on, and reasons for, complementary feeding practices, such as providing sweet beverages or unhealthy foods, were not explored. Further qualitative studies should explore these issues in more depth. Third, the study is cross-sectional, meaning we cannot ascertain causal relationships. Forth, all data on complementary feeding were collected using the 24-h recall which may not reflect children’s diets over a longer period. There is a possibility that some respondents may provide answers that they think will be favoured by others. This presumes that they know what the preferred practice is. Last, the sample of children cared for by people who were not their mothers may be relatively small especially when they are separated into different sub-categories, such as child age, and caregiver age.

## 5. Conclusions

This study points out the sociodemographic and economic factors of inappropriate complementary feeding practices among Thai children aged 6–23 months according to the newly developed WHO IYCF indicators. Overall, our study demonstrated that the child’s age and the characteristics of primary caregivers, including non-mother caregivers, were crucial determinants of complementary feeding practice. Although most children aged 6–8 months were introduced solid, semi-solid, or soft foods, more than half of them still have challenges meeting MDD and MAD. Moreover, egg/flesh food and fruit and vegetable consumption increased with child’s age. Consequently, younger children, especially those cared for by non-mothers often miss essential dietary components. While older children are likely to achieve MDD, the higher percentage of them consumed SwB and UFC when compared to those 6–8 months of age. Moreover, dietary diversity was a positive attribute for children of middle-class mothers, this group also had high sweet beverage consumption, indicating diversity alone may not be only a positive factor for diets. Our findings can be applied to develop or update complementary feeding guidelines to educate mothers about appropriate feeding practices. Guidelines may need to specifically address non-mother caregivers. Furthermore, stakeholders can employ the results to tailor-made nutritional programs and interventions addressing inappropriate complementary feeding among different groups of children.

## Figures and Tables

**Table 1 nutrients-14-04370-t001:** WHO complementary feeding indicators *.

Indicator	Short Name	Age Group	Definition
Introduction of solid, semi-solid or soft foods 6–8 months	ISSF	Infants 6–8 months of age	Percentage of infants 6–8 months of age who consumed solid, semi-solid or soft foods during the previous day.
Minimum dietary diversity 6–23 months	MDD	Children 6–23 months of age	Percentage of children 6–23 months of age who consumed foods and beverages from at least five out of eight defined food groups during the previous day.
Minimum meal frequency 6–23 months	MMF	Children 6–23 months of age	Percentage of children 6–23 months of age who consumed solid, semi-solid or soft foods (but also including milk feeds for non-breastfed children) the minimum number of times or more during the previous day. The minimum number of times is defined as: two feedings of solid, semi-solid or soft foods for breastfed infants aged 6–8 months; three feedings of solid, semi-solid or soft foods for breastfed children aged 9–23 months; four feedings of solid, semi-solid or soft foods or milk feeds for non-breastfed children aged 6–23 months whereby at least one of the four feeds must be a solid, semi-solid or soft feed.
Minimum milk feeding frequency for non-breastfed children 6–23 months	MMFF	Children 6–23 months of age	Percentage of non-breastfed children 6–23 months of age who consumed at least two milk feeds during the previous day.
Minimum acceptable diet 6–23 months	MAD	Children 6–23 months of age	Percentage of children 6–23 months of age who consumed a minimum acceptable diet during the previous day. The minimum acceptable diet is defined as: for breastfed children: receiving at least the minimum dietary diversity and minimum meal frequency for their age during the previous day; for non-breastfed children: receiving at least the minimum dietary diversity and minimum meal frequency for their age during the previous day as well as at least two milk feeds.
Egg and/or flesh food consumption 6–23 months	EFF	Children 6–23 months of age	Percentage of children 6–23 months of age who consumed egg and/or flesh food during the previous day.
Sweet beverage consumption 6–23 months	SwB	Children 6–23 months of age	Percentage of children 6–23 months of age who consumed a sweet beverage during the previous day.
Unhealthy food consumption 6–23 months	UFC	Children 6–23 months of age	Percentage of children 6–23 months of age who consumed selected sentinel unhealthy foods (such as fried foods, confections) during the previous day.
Zero vegetable or fruit consumption 6–23 months	ZVF	Children 6–23 months of age	Percentage of children 6–23 months of age who did not consume any vegetables or fruits during the previous day.

* World Health Organization. (2021). Indicators for assessing infant and young child feeding practices: definitions and measurement methods.

**Table 2 nutrients-14-04370-t002:** Percentage of individual-, caregiver-, household-, and community-level characteristics of children aged 6–23 months, Thailand 2019 (*n* = 4125).

Characteristics	All (*n* = 4125)	Children Provide Care by		Characteristics	All (*n* = 4125)	Children Provide Care by	
		Mother (*n* = 3152)	Other (*n* = 973)			Mother (*n* = 3152)	Other (*n* = 973)
Child characteristics				Household characteristics			
Male	51.49	51.33	52.00	Household wealth index			
Currently breastfed *	29.86	37.74	2.91	Poorest	23.22	20.69	31.45
Age (months)				Second	22.52	20.69	28.47
6–8	13.16	14.34	9.35	Middle	21.87	22.27	20.55
9–11	17.04	17.7	14.90	Fourth	18.30	19.67	13.87
12–17	32.15	32.07	32.37	Richest	14.08	16.69	5.65
18–23	37.65	35.88	43.37	Community level characteristics			
Maternal/caregiver characteristics				Residence			
Age (years)				Urban	32.70	35.50	23.64
15–19	26.81	20.86	4.46	Geographical region			
20–35	66.61	72.02		Bangkok and central	30.64	32.87	23.43
>35	6.58	7.12	95.54	North	14.06	14.82	11.60
Education				Northeast	32.99	26.30	54.68
Kindergarten and primary	18.13	13.43	75.40	South	22.30	26.02	10.28
High school and above	81.87	86.57	24.60				
Primary language							
Thai	72.41	86.64	26.31				

* reported receiving any breastmilk in the past 24 h.

**Table 3 nutrients-14-04370-t003:** Percentage of children who met the complementary feeding indicators according to their age range and primary caregivers (*n* = 4125).

Indicators	Child Age	All (*n* = 4125)	Primary Caregiver		*p*-Value *
			Mother (*n* = 3152)	Other (*n* = 973)	
Introduction of solid, semi-solid or soft foods 6–8 months (ISSSF)	6–8 months	92.00%	91.56%	93.41%	0.061
Minimum dietary diversity (MDD)	6–8 months	42.42%	45.35%	32.97%	<0.001
	9–11 months	64.87%	65.95%	61.38%	0.009
	12–17 months	73.99%	75.96%	67.62%	<0.001
	18–23 months	72.53%	74.09%	67.54%	<0.001
Minimum meal frequency (MMF)	6–8 months	88.80%	86.48%	96.34%	<0.001
	9–11 months	96.10%	96.32%	95.42%	0.184
	12–17 months	97.28%	96.57%	99.65%	<0.001
	18–23 months	95.64%	95.61%	95.68%	0.934
Minimum milk feeding frequency for non-breastfed children (MMFF)	6–8 months	95.64%	96.61%	93.98%	<0.001
	9–11 months	97.96%	99.29%	93.65%	<0.001
	12–17 months	97.18%	97.09%	97.55%	0.456
	18–23 months	93.50%	93.48%	93.57%	0.947
Minimum acceptable diet (MAD)	6–8 months	38.67%	42.19%	27.27%	<0.001
	9–11 months	62.93%	64.31%	58.45%	0.001
	12–17 months	69.14%	70.39%	65.10%	0.002
	18–23 months	68.80%	70.37%	63.68%	<0.001
Egg and/or flesh food consumption (EFF)	6–8 months	70.21%	72.54%	62.64%	<0.001
	9–11 months	88.85%	89.89%	85.52%	<0.001
	12–17 months	94.64%	95.15%	93.02%	0.010
	18–23 months	96.41%	96.61%	95.79%	0.230
Sweet Beverage consumption (SwB)	6–8 months	67.98%	62.17%	86.81%	<0.001
	9–11 months	72.65%	68.28%	86.82%	<0.001
	12–17 months	74.01%	69.90%	87.30%	<0.001
	18–23 months	70.81%	70.60%	71.53%	0.572
Unhealthy food consumption (UFC)	6–8 months	52.72%	50.00%	61.54%	<0.001
	9–11 months	65.09%	62.66%	72.92%	<0.001
	12–17 months	72.24%	71.61%	74.29%	0.100
	18–23 months	78.04%	77.23%	80.66%	0.023
Zero vegetable or fruit consumption (ZVF)	6–8 months	35.18%	33.48%	40.66%	<0.001
	9–11 months	16.90%	14.18%	25.69%	<0.001
	12–17 months	13.79%	12.67%	17.46%	<0.001
	18–23 months	12.65%	11.43%	16.58%	<0.001

* Chi-square test.

**Table 4 nutrients-14-04370-t004:** Factors associated with complementary feeding indicators, ISSF, MDD, MMF, MMFF, MAD, EFF, the multivariate logistic regression analysis.

	ISSSF ^1^		MDD ^2^		MMF ^3^		MMFF ^4^		MAD ^5^		EFF ^6^	
	AOR (95%CI)											
	Primary Caregiver		Primary Caregiver		Primary Caregiver		Primary Caregiver		Primary Caregiver		Primary Caregiver	
	Mother	Other	Mother	Other	Mother	Other	Mother	Other	Mother	Other	Mother	Other
Child gender (Reference category male)												
Female	0.67 (0.33–1.37)	5.95 (0.35–101.68)	1.04 (0.89–1.23)	0.81 (0.45–1.46)	0.85 (0.60–1.21)	0.62 (0.14–2.85)	0.86 (0.54–1.39)	0.62 (0.16–2.46)	1.04 (0.8–1.23)	1.10 (0.21–5.62)	0.87 (0.65–1.15)	0.76 (0.25–2.35)
Child age (months) (Reference category 6–8 months)												
9–11	-	-	2.35 ** (1.80–3.07)	3.01 (0.95–9.50)	3.54 ** (2.06–6.07)	0.22 (0.01–6.20)	4.54 (0.90–22.86)	1.87 (0.10–35.98)	2.40 ** (1.84–3.13)	3.28 * (1.00–10.76)	3.35 ** (2.32–4.85)	6.90 * (1.62–29.31)
12–17	-	-	4.35 ** (3.38–5.60)	4.75 ** (1.66–13.62)	3.03 ** (1.86–4.93)	N/A	1.43 (0.54–3.76)	2.72 (0.21–34.72)	3.52 ** (2.74–4.51)	5.61 ** (1.90–16.59)	7.18 ** (4.89–10.54)	18.95 ** (4.28–83.95)
18–23	-	-	3.83 ** (2.99–4.92)	3.25 * (1.16–9.10)	1.53 (0.98–2.39)	0.11 (0.00–2.47)	0.52 (0.22–1.24)	0.38 (0.04–3.40)	3.25 ** (2.56–4.20)	3.59 * (1.24–10.39)	9.54 ** (6.22–14.61)	84.02 ** (8.64–817.36)
Mother’s/caregiver’s age (years) (Reference category > 35 years)												
15–19	1.70 (0.18–16.02)	N/A	1.05 (0.73–1.51)	1.65 (0.34–7.94)	1.02 (0.47–2.22)	0.19 (0.01–3.81)	0.55 (0.19–1.62)	0.07 * (0.01–0.75)	1.01 (0.71–1.45)	0.91 (0.19–4.33)	1.03 (0.55–1.96)	0.67 (0.06–7.02)
20–35	0.50 (0.18–1.38)	N/A	0.87 (0.71–1.08)		0.91 (0.59–1.40)		0.54 (0.26–1.12)		0.85 (0.69–1.05)		0.83 (0.57–1.21)	
Mother’s/caregiver’s language (Reference category Thai)												
Non-Thai	2.65 (0.54–13.01)	1.20 (0.05–29.39)	1.24 (0.92–1.67)	1.40 (0.28–7.07)	0.79 (0.88–2.34)	1.71 (0.02–128.34)	0.67 (0.31–1.44)	0.34 (0.02–6.58)	1.16 (0.87–1.54)	1.10 (0.21–5.62)	2.87 ** (1.54–5.35)	0.18 (0.01–2.19)
Mother’s/caregiver’s education (Reference category Kindergarten and Primary)												
≥High school	1.58 (0.57–4.37)	N/A	1.41 * (1.10–1.81)	0.93 (0.45–1.91)	1.44 (0.88–2.34)	0.63 (0.10–4.00)	1.30 (0.65–2.62)	1.29 (0.21–7.88)	1.41 * (1.10–1.81))	1.09 (0.53–2.25)	1.76 * (1.18–2.62)	1.58 (0.39–6.44)
Household wealth index (Reference category poorest)												
Second	2.48 (0.83–7.44)	4.63 (0.16–132.80)	1.42 * (1.11–1.84)	2.74 * (1.29–5.79)	1.05 (0.61–1.80)	6.94 (0.68–71.02)	1.36 (0.67–2.76)	4.03 (0.77–20.98)	1.40 * (1.09–1.80)	2.95 * (1.39–6.27)	1.25 (0.81–1.94)	0.72 (0.19–2.72)
Middle	3.17 (0.97–10.33)	5.26 (0.22–123.14)	1.41 * (1.09–1.83)	3.08 * (1.29–7.40)	0.97 (0.55–1.71)	2.35 (0.27–20.34)	1.22 (0.61–2.45)	3.96 (0.41–38.61)	1.36 * (1.05–1.75)	3.32 * (1.38–8.03)	1.16 (0.74–1.80)	1.32 (0.28–6.31)
Fourth	1.09 (0.43–2.78)	N/A	1.62 ** (1.24–2.13)	0.06 (0.96–7.30)	0.55 * (0.31–0.96)	2.36 (0.17–33.21)	1.85 (0.85–4.01)	7.18 (0.61–84.86)	1.47 * (1.12–1.92)	2.93 * (1.06–8.12)	1.25 (0.78–1.99)	6.22 (0.46–83.43)
Richest	0.82 (0.32–2.10)	0.03 (0.00–1.24)	2.20 ** (1.63–2.97)	N/A	0.78 (0.41–1.48)	N/A	3.47 * (1.28–9.35)	N/A	2.09 ** (1.56–2.80)	N/A	1.60 (0.96–2.69)	N/A
Residence (Reference category urban)												
Rural	0.94 (0.43–2.05)	35.18 * (1.51–822.10)	1.06 (0.88–1.26)	0.89 (0.42–1.88)	1.09 (0.75–1.58)	0.34 (0.02–4.83)	0.94 (0.55–1.61)	0.82 (0.14–4.64)	1.08 (0.91–1.29)	0.81 (0.38–1.73)	1.08 (0.80–1.46)	0.84 (0.19–3.79)
Geographical region (Reference category Bangkok and central)												
North	1.17 (0.38–3.63)	0.04 (0.00–1.74)	0.66 ** (0.51–0.86)	0.18 * (0.05–0.64)	1.24 (0.69–2.21))	3.04 (0.09–102.21)	0.63 (0.31–1.26)	0.47 (0.07–3.11)	0.61 ** (0.48–0.79)	0.21 * (0.06–0.74)	1.02 (0.67–1.55)	0.10 * (0.02–0.65)
Northeast	1.20 (0.46–3.13)	7.47 (0.22–251.49)	1.10 (0.88–1.36)	2.08 (0.99–4.34)	0.95 (0.60–1.49)	1.96 (0.33–11.79)	1.10 (0.57–2.11)	4.44 (0.84–23.38)	1.10 (0.88–1.36)	2.25 * (0.04–1.06)	2.20 ** (1.50–3.24)	0.82 (0.22–3.15)
South	1.00 (0.37—2.67)	0.24 (0.00–11.63)	1.27 * (1.00–1.61)	0.60 (0.41–4.60)	1.40 (0.84–2.32)	N/A	1.06 (0.53–2.11)	N/A	1.23 (0.98–1.55)	1.43 (0.43–4.84)	1.85 ** (1.24–2.76)	N/A
Currently breastfeeding *** (Reference category No)												
Yes	0.26 ** (0.11–0.63)	27.52 (0.09–8842.30)	1.35 ** (1.13–1.62))	2.14 (0.25–17.93)	0.10 ** (0.06–0.16)	0.01 * (0.00–0.40)	-	-	0.98 (0.82–1.17)	0.33 (0.02–5.64)	0.50 ** (0.37–0.67)	0.38 (0.03–5.43)

* *p* < 0.05; ** *p* < 0.01, N/A = not available; *** reported receiving any breastmilk in the past 24 h; ^1^ ISSF = Introduction of solid, semi-solid or soft foods 6–8 months; ^2^ MDD = Minimum dietary diversity 6–23 months; ^3^ MMF = Minimum meal frequency 6–23 months; ^4^ MMFF = Minimum milk feeding frequency for non-breastfed children 6–23 months; ^5^ MAD = Minimum acceptable diet 6–23 months; ^6^ EFF = Egg and/or flesh food consumption 6–23 months; AOR = Adjusted odds ratio; 95%CI = 95% confidence interval.

**Table 5 nutrients-14-04370-t005:** Factors associated with complementary feeding indicators, SwB, UFC, ZVF, the multivariate logistic regression analysis.

	SwB ^1^		UFC ^2^		ZVF ^3^	
	AOR (95%CI)					
	Primary Caregiver		Primary Caregiver		Primary Caregiver	
	Mother	Other	Mother	Other	Mother	Other
Child gender (Reference category male)						
Female	1.03 (0.87–1.23)	1.87 (0.81–4.32)	1.10 (0.88–1.38)	1.47 (0.57–3.83)	1.00 (0.82–1.24)	1.41 (0.70–2.83)
Child age (months) (Reference category 6–8 months)						
9–11	1.12 (0.84–1.50)	2.21 (0.35–13.74)	1.87 * (1.15–3.04)	0.47 (0.03–7.40)	0.34 ** (0.25–0.47)	0.80 (0.24–2.67)
12–17	0.97 (0.74–1.26)	2.75 (0.58–13.11)	2.72 ** (1.75–4.23)	0.40 (0.03–4.79)	0.29 ** (0.22–0.39)	0.43 (0.14–1.30)
18–23	0.78 (0.59–1.02)	0.38 (0.09–1.52)	3.44 ** (2.20–5.37)	2.07 (0.16–26.62)	0.27 ** (0.20–0.36)	0.50 (0.17–1.51)
Mother’s/caregiver’s age (years) (Reference category > 35 years)						
15–19	0.92 (0.63–1.34)	0.19 (0.03–1.06)	0.86 (0.53–1.40)	8.73 (0.48–157.89)	0.92 (0.59–1.43)	2.10 (0.37–11.90)
20–35	0.94 (0.76–1.17)		1.01 (0.76–1.36)		1.00 (0.77–1.30)	
Mother’s/caregiver’s language (Reference category Thai)						
Non-Thai	1.08 (0.80–1.45)	0.64 (0.08–5.10)	0.79 (0.54–1.16)	0.50 (0.06–4.41)	0.67 * (0.45–0.98)	0.52 (0.20–1.34)
Mother’s/caregiver’s education (Reference category Kindergarten and Primary)						
≥ High school	1.27 (0.98–1.65)	1.44 (0.54–3.81)	1.16 (0.82–1.62)	0.16 * (0.05–0.58)	0.73 * (0.54–0.98)	0.52 (0.20–1.34)
Household wealth index (Reference category poorest)						
Second	1.06 (0.82–1.38)	0.45 (0.14–1.46)	0.87 (0.60–1.28)	6.24 * (1.50–25.89)	0.77 (0.57–1.04)	0.72 (0.32–1.65)
Middle	1.29 (0.98–1.69)	0.27 * (0.08–0.91)	0.66 * (0.46–0.96)	2.81 (0.71–11.07)	0.67 * (0.49–0.92)	0.51 (0.19–1.41)
Fourth	1.69 ** (1.25–2.27)	0.56 (0.13–2.46)	0.54 ** (0.36–0.80)	0.83 (0.20–3.49)	0.54 ** (0.38–0.76)	0.18 (0.04–0.76)
Richest	1.18 (0.87–1.60)	3.11 (0.06–172.67)	0.48 ** (0.31–0.73)	3.68 (0.12–113.02)	0.45 ** (0.31–0.66)	N/A
Residence (Reference category urban)						
Rural	0.92 (0.76–1.10)	1.89 (0.74–4.79)	1.19 (0.92–1.52)	1.34 (0.37–4.83)	1.03 (0.82–1.29)	2.03 (0.76–5.42)
Geographical region (Reference category Bangkok and central)						
North	0.75 * (0.57–0.99)	1.89 (0.74–4.79)	0.73 (0.49–1.06)	0.42 (0.06–3.10)	1.10 (0.79–1.54)	2.74 (0.80–9.38)
Northeast	0.96 (0.76–1.21)	0.44 (0.10–1.92)	0.97 (0.69–1.34)	0.16 * (0.03–0.74)	1.11 (0.85–1.46)	0.51 (0.21–1.22)
South	1.00 (0.78–1.27)	1.63 (0.58–0.29)	0.43 ** (0.32–0.58)	0.21 (0.03–1.30)	0.89 (0.66–1.21)	0.43 (0.08–2.57)
Currently breastfeeding *** (Reference category No)						
Yes	0.16 ** (0.14–0.20)	0.02 ** (0.00–0.23)	0.08 (0.68–1.64)	4.68 (0.18–119.32)	1.04 (0.84–1.30)	0.46 (0.04–5.56)

* *p* < 0.05; ** *p* < 0.01; *** reported receiving any breastmilk in the past 24 h; ^1^ SwB = Sweet beverage consumption 6–23 months; ^2^ UFC = Unhealthy food consumption 6–23 months; ^3^ ZVF = Zero vegetable or fruit consumption 6–23 months; AOR = Adjusted odds ratio; 95%CI = 95% confidence interval.

## Data Availability

Not applicable.

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
