# Peer review of "Determinants of Complementary Feeding Indicators: A Secondary Analysis of Thailand Multiple Indicators Cluster Survey 2019"

_nutrients, 2022, doi:10.3390/nu14204370_

Round 1

Reviewer 1 Report

This is a nicely written fair study which gives a piece of knowledge about children feeding in Thailand. It can be very helpful to apply it in some meta-analysis in the future or it can be very helpful for the stakeholders in Thailand and other countries from this region. I have no remarks and from my perspective it can be published as it is.

Author Response

Thank you for your comments. We appreciate you for taking the time to review our manuscript.

Reviewer 2 Report

The authors have obtained interesting results about the analysis of a nationally representative survey of Thai children, mothers and other caregivers reveals mixed patterns of both appropriate and inappropriate child feeding  practices with some indicators at concerning levels. Overall, the complementary feeding practices of other caregivers were less favourable than those of mothers and children of mothers with higher education and from wealthier households had better appropriate complementary feeding.  The findings from this study provide further support for associations between household income level and complementary feeding indicators on minimum dietary diversity (MDD) and minimum acceptable diet (MAD). The authors conclude that nutrition programs addressing different feeding problems should be developed specifically for different primary caregiver and demographic groups.

Please, consider these suggestions:

Abstract

The authors should specify the study N and age children.

Line 22:    They should indicate age older children.
Introduction
Lines 78-79: The authors should remove the acronyms (IYCF), this one is not repeated.

Material and Methods
Line 105: The authors should remove the acronyms (SDGs), this one is not repeated.
I agree with the authors that in a limitation of the study “all data on complementary feeding were collected using the 24-hour recall which may not reflect children's diets over a longer period”. A food consumption frequency survey or reminder of three non-consecutive days and one of them a holiday would have been interesting.

Conclusions
The authors have the next aim: “ The aim is  use the newly developed WHO infant feeding indicators to assess patterns of complementary feeding among Thai children 6-23 months of age, the distribution of these patterns, and to identify the potential risk factors associated with inappropriate complementary feeding practices in Thailand”.
The conclusions must respond to the stated aim and indicate the study population.

Author Response

Abstract

1. The authors should specify the study N and age children.

Response: • Thank you for pointing this out. We have added N and age of children as your suggestion: “This study uses data from the Thailand Multiple Indicators Survey 2019, to identify the determi-nants of CF practices among 6-23-month children (N= 4,125) using the newly developed WHO indicators” (Line 19-20)
2. Line 22: They should indicate age older children.

Response: • Thank you for your suggestion. We have added the suggested content to the manuscript: “Older children aged 9-23 months, not only have better minimum dietary diversity (MDD), minimum acceptable diet (MAD), and egg and/or flesh food consumption (EFF), but also tend to consume more unhealthy foods.” (Line 23)

Introduction

3. Lines 78-79: The authors should remove the acronyms (IYCF), this one is not repeated.

Response: • Thank you for pointing this out. We have removed the acronyms (IYCF) (Line 80)
Material and Methods

4. Line 105: The authors should remove the acronyms (SDGs), this one is not repeated. I agree with the authors that in a limitation of the study “all data on complementary feeding were collected using the 24-hour recall which may not reflect children's diets over a longer period”. A food consumption frequency survey or reminder of three non-consecutive days and one of them a holiday would have been interesting.

Response: • Thank you for pointing this out. We have removed the acronyms (SDGs) (Line 106)
Conclusions

5. The authors have the next aim: “The aim is use the newly developed WHO infant feeding indicators to assess patterns of complementary feeding among Thai children 6-23 months of age, the distribution of these patterns, and to identify the potential risk factors associated with inappropriate complementary feeding practices in Thailand”. The conclusions must respond to the stated aim and indicate the study population.

Response: • We appreciated the reviewer’s helpful comments, and have rewritten and added the suggested content to the manuscript: “This study points out the sociodemographic and economic factors of inappropriate complementary feeding practices among Thai children aged 6-23 months according to the newly developed WHO IYCF indicators. Overall, our study demonstrated that the child’s age and the characteristics of primary caregivers, including non-mother caregivers, were crucial determinants of complementary feeding practice. Although most children aged 6-8 months were introduced solid, semi-solid, or soft foods, more than half of them still have challenges meeting MDD and MAD. Moreover, egg/ flesh food and fruit and vegetable consumption increased with the child’s age. Consequently, younger children, especially those cared for by non-mothers, often miss essential dietary components. While older children are likely to achieve MDD, the higher percentage of them consumed SwB and UFC when compared to those 6-8 months of age. Moreover, dietary diversity was a positive attribute for children of middle-class mothers, this group also had high sweet beverage consumption, indicating diversity alone may not be only a positive factor for diets. Our findings can be applied to develop or update complementary feeding guidelines to educate mothers about appropriate feeding practices. Guidelines may need to specifically address non-mother caregivers. Furthermore, stakeholders can employ the results to tailor-made nutritional programs and interventions addressing inappropriate complementary feeding among different groups of children.” (Line 349-366)